# Evaluation of Risk Factors Influencing Tick-Borne Infections in Livestock Through Molecular Analyses

**DOI:** 10.3390/microorganisms13010139

**Published:** 2025-01-11

**Authors:** Lorena Cardillo, Claudio de Martinis, Giovanni Sgroi, Alessia Pucciarelli, Gerardo Picazio, Maurizio Viscardi, Luisa Marati, Maria Ottaiano, Roberta Pellicanò, Nicola D’Alessio, Vincenzo Veneziano, Giovanna Fusco

**Affiliations:** 1Istituto Zooprofilattico Sperimentale del Mezzogiorno, Department of Animal Health, 80055 Portici, Italy; lorena.cardillo@izsmportici.it (L.C.); giovanni.sgroi@izsmportici.it (G.S.); alessia.pucciarelli@izsmportici.it (A.P.); gerardo.picazio@izsmportici.it (G.P.); maurizio.viscardi@izsmportici.it (M.V.); luisa.marati@izsmportici.it (L.M.); nicola.dalessio@izsmportici.it (N.D.); giovanna.fusco@izsmportici.it (G.F.); 2Istituto Zooprofilattico Sperimentale del Mezzogiorno, Department of Epidemiology and Biostatistics Regional Observatory (OREB), 80055 Naples, Italy; maria.ottaiano@izsmportici.it (M.O.); roberta.pellicano@izsmportici.it (R.P.); 3Department of Veterinary Medicine and Animal Productions, Università degli Studi di Napoli Federico II, 80138 Naples, Italy; vinvene@unina.it

**Keywords:** vector-borne diseases, tick-borne diseases, livestock, real-time PCR, zoonosis

## Abstract

Climate changes and human-related activities are identified as major factors responsible for the increasing distribution and abundance of vectors worldwide and, consequently, of vector-borne diseases (VBDs). Farmed animals, during grazing or in establishments with the absence of biosecurity measures, can easily be exposed to wildlife showing high-risk of contagion of several infectious diseases, including VBDs. Furthermore, livestock represents an interface between wildlife and humans, and thus, promoting the transmission pathway of VBDs. Little is known about the presence and prevalence of VBDs in livestock in Southern Italy; therefore, the present study evaluated the circulation of zoonotic VBDs in livestock and potential risk of exposure. A total of 621 whole blood samples belonging to cattle and buffaloes (*n* = 345) and small ruminants (*n* = 276) were examined by molecular examinations for the detection of tick-borne pathogens (TBPs). High prevalence (66.3%) for at least one agent was observed. Moreover, the risk of exposure related to environmental features was assessed, as follows: presence of humid areas, high-density of animals, and sample collection during May. These results show a high circulation of TBPs among livestock and underline the need for surveillance in high-risk habitats for public health.

## 1. Introduction

Vector-borne diseases (VBDs) are caused by a wide range of infectious and parasitic agents transmitted by blood-feeding arthropods, such as ticks, fleas, lice, kissing bugs, mosquitoes, and sand flies [1]. Vector-borne pathogens (VBPs) are responsible for 17% of all infectious diseases accounting for more than 700,000 deaths annually [2]. Thereby, VBPs represent a global threat in public health, especially due to the emerging/re-emerging trend of certain zoonotic agents, including *Anaplasma phagocytophilum*, *Babesia*/*Theileria* spp., *Borrelia* spp., *Coxiella burnetii*, Crimean–Congo hemorrhagic fever virus (CCHV), *Ehrlichia* spp., spotted fever group rickettsiae, and tick-borne encephalitis virus (TBEV) [3,4]. Vector-borne diseases are strictly related to the presence and ecology of the competent vectors in a determined area, which, in association with the biology of the pathogen, determine the geographical distribution of the diseases [5]. Climate features largely influence the distribution and abundance of vectors. Indeed, while in developing countries, the main VBDs are represented by mosquito-borne diseases mostly connected to poor living conditions [6], in the industrialized countries of the northern regions, tick-borne diseases are of major concern [7]. Several factors are responsible for the emergence and distribution of TBDs, in particular, the abundance of ticks, presence and prevalence of TBPs, and the exposure to tick bites [7]. Human-related environmental changes, such as agricultural strategies, wildlife management, deforestation, and global warming, are strongly involved in the alteration of the ecosystems, possibly affecting the arthropod–host interaction and circulation of TBPs [5,8] and driving the rise of emerging infectious diseases [9]. In addition, the increasing density of synanthropic species around Europe may promote the spatial overlap between wildlife and farmed animals, and thus, promoting the transmission pathway of VBPs between animals and humans in the livestock industry [10]. This aspect was recently enhanced during the COVID-19 pandemic, when the reduced outdoor human activity induced several wildlife species to move into close proximity to the urban, peri-urban, and farmland settlements of Europe [11,12], and thus increasing the risk of contagion both for humans and livestock. Indeed, as observed for *Anaplasma phagocytophilum*, the agent responsible for anaplasmosis in livestock, the presence of the pathogen is strictly linked to the presence of specific wildlife species (red deer). Thus, the presence of the main transmission host is a predictive factor of anaplasmosis in livestock [5].

*A. phagocytophilum* infects small ruminants, domestic, and wild animals, causing reduced milk production in cattle human granulocytic anaplasmosis in humans. Different studies have been conducted on the presence of *A. phagocytophilum* in Italy both in the tick vectors and in the wild and domestic reservoirs. The prevalence of *A. phagocytophilum* embraces the whole Italian territory from the Alps to the southern and insular regions [13]. At the same time, a high prevalence of *Ehrlichia canis* (21%) has been identified in abortion products from the sheep and goats of Sardinia Island [14]. Q fever, caused by *Coxiella burnetii*, is also a zoonotic tick-borne disease that mainly affects sheep, goats, buffaloes, and cattle with a mostly asymptomatic course, although it can be responsible for infertility, stillbirth, placental retention, and abortions. Ticks are an important vector and reservoir of the bacterium that is transmitted transovarially to the offspring. In Italy, recent epidemiological studies conducted in dairy cattle farms have demonstrated a high prevalence of Q fever in the north-western regions [15]. In addition, species of *Babesia* (e.g., *Babesia bigemina*, *Babesia bovis*, *Babesia divergens*) [16] and *Borrelia* spp. (e.g., *Borrelia theileri*) [17] are responsible for babesiosis and borrelliosis in cattle, respectively. Although CCHV is uncommon compared to other tick borne agents, a study identified a high seroprevalence of this virus in bovines in southern Italy, whereas TBE appears to be restricted to the central and north-eastern part of the country [18,19].

Thereby, TBP infection surveillance in livestock industry has a crucial role in preserving animal welfare, food safety, and animal productivity [5,17], as well as public health. Indeed, a recent epidemiological survey among outdoor workers, such as farmers, forestry workers, veterinarians, and geologists/agronomists, revealed a high seroprevalence of zoonotic TBPs in farmers, in particular 30.3% for *C. burnetii*, 15.3% for *Rickettsia conorii*, 8.8% for *Bartonella henselae,* and 4.1% for *Borrelia* spp. Altogether, these data highlight the need for a proper public health response to VBPs in high-risk areas and workplaces [20].

Therefore, in the present study, the aim was to investigate the exposure of livestock, sheep (*Ovis aries*), goat (*Capra hircus*), cattle (*Bos Taurus*), and Italian Mediterranean buffalo (*Bubalus bubalis*) to selected TBPs and related environmental risk-factors of infection in the livestock industry of southern Italy.

## 2. Materials and Methods

### 2.1. Study Area and Sampling

A total of 621 whole blood samples in EDTA, belonging to 310 buffaloes, 35 cattle, 249 sheep, and 27 goats, were examined. The samples were collected by the veterinary local health authorities during routine activities for the provisions of the national surveillance plans. The animals belonged to 126 different farms located in the five provinces of the Campania region, as follows: Caserta, Avellino, Benevento, Salerno, and Naples. The number of the tested animals and farms was selected proportionally to the registered establishments of the territories and on the basis of climatic and hydro-geographic features that may impact on vector presence and abundance according to what is reported in the bibliography regarding the predilection of the vector for such environments [21].

All the examined blood samples were obtained by the local veterinary health authority for national and regional eradication and control plans. The Istituto Zooprofilattico Sperimentale del Mezzogiorno is the official laboratory designated by the Italian Ministry of Health; therefore, according to the national regulations and internal policies, ethical approval was deemed unnecessary.

### 2.2. Nucleic Acid Extraction and Real Time Polymerase Chain Reaction (PCR) Protocols

Aliquots of 200 µL whole blood samples were collected and underwent nucleic acid extraction and purification with QIAsymphony automated system (Qiagen, Hilden, Germany) using QIAsymphony DSP Virus/Pathogen Mini Kit (Qiagen) following the manufacturer’s instructions, eluted in 60 µL and stored at −80 °C until use. Each extraction session was performed in the presence of an extraction control, composed of nuclease-free water. Elutes were submitted to real time PCR for the detection of tick-borne pathogens, using specific primer sets and probes, and heat profiles, as reported in Table 1, in the presence of specific positive and negative controls.

### 2.3. Statistical Analysis

The possible association between the outcome (presence/absence of infection in the factories) and the environmental variables that may be responsible for exposure to infectious agents was evaluated. In particular, the altitude of the territory, the presence of wetlands and pastures, and months of the year were analyzed. Fisher’s exact test was used for the qualitative variables and a Wilcox test for the quantitative variables. The presence of the infection was established with the detection of at least one positive animal in the farm. Results were considered as statistically significant with a *p* value < 0.05.

Statistical analysis was conducted using R Studio software, version 4.0.2 by the Epidemiology and Biostatistics Regional Observatory (OREB).

## 3. Results

A total of 126 livestock establishments were investigated for the presence of TBPs. In particular, 121 (96%) were intended for housing sheep or goats, and 5 (4%) for cattle or buffaloes. Out of the 126 farms, 41 (32.5%, 95% CI: 25.0–41.1) tested positive for at least one TBP, being 36 (28.6%, 95% CI: 21.4–37.0) and 5 (3.9%, 95% CI: 1.7–8.9) from sheep/goat and cattle/buffalo establishments, respectively. The distribution of the tested positive farms of the study area, including all co-infection cases, is shown in Figure 1.

A total of 310 buffalo, 35 cattle, 249 sheep, and 27 goats were examined. On a total of 621 animals examined, 412 (66.3%, 95% CI: 62.5–69.9) scored positive for at least one TBP. In particular, 240 (38.6%, 95% CI: 34.9–42.5) tested positive for *Anaplasma phagocytophilum* (118 buffalo, 12 cattle, 82 sheep, and 28 goats) followed by 143 (23.0%, 95% CI: 19.9–26.5) to *Babesia* spp. (79 buffalo, 7 cattle, 55 sheep, and 2 goats). In addition, 26 animals (4.2%, 95% CI: 2.9–6.1) were found to be positive for *Borrelia burgdorferi* sensu lato complex (21 buffalo, 3 cattle, and 2 sheep) and only 3 animals (0.5%, 95% CI: 0.2–1.4) tested positive for *Coxiella burnetii* (2 buffalo, and 1 sheep). No animal herein screened showed positive results for the TBE virus and CCHFV. Data on the specific pathogen according to the host species are detailed in Table 2. Any interpretation of such data sets should be treated with caution, as these results will be susceptible to the same biases that are associated with any PCR protocol; in order to avoid this, we addressed real-time PCR protocols with a specific primer set.

The statistical analyses showed a significant difference (*p* < 0.05) in the positivity of farms to at least one pathogen according to the following examined variables: (i) field context, (ii) number of animals housed per farm, and (iii) collection period of the year. In particular, farms located in humid areas, housing a high number of animals, along with the sample collection run during May, resulted in representing the main features related to TBP infections. Details of the statistical analyses are reported in Table 3. None of the positive animals showed clinical signs ascribable to the investigated infections.

## 4. Discussion

Vector-borne diseases represent a global threat, being responsible for 17% of all the infectious diseases worldwide [2]. Ticks are competent vectors of numerous bacteria, viruses, and parasites, being the second most significant vector after mosquitos for the transmission of VBDs. Currently, several studies have demonstrated a substantial geographic expansion of tick distribution and, consequently, of TBDs, even in areas that were previously considered as free or where the environmental conditions were deemed not favorable for the survival and development of these arthropods [28,29]. This phenomenon has been associated with the trends in the climate, as ticks are strictly temperature- and humidity-dependent. Therefore, ticks are spreading to higher latitudes and altitudes, as well as developing a prolonged season of activity, useful for the transmission and development of pathogens [29,30,31]. In a 2010 joint report by the European Centre for Disease Prevention and Control (ECDC) and the European Food Safety Authority (EFSA) on the distribution of ticks and TBDs in Europe and the Mediterranean basin, 14 tick-borne zoonoses were reviewed. It was highlighted that in addition to the climate changes, the movement of migratory birds which can transport and spread biological agents, the techniques adopted in animal husbandry, the wild and exotic species introduced into the environment, and the movements of the human population played a key role in the distribution and spread of ticks [32]. In Italy, the TBDs are of great epidemiological significance, mainly Lyme borreliosis, rickettsiosis, tick-borne relapsing fever, tularemia, ehrlichiosis, and TBE [33]. Indeed, some studies conducted in Italy among human biting ticks and related TBDs reported that the overall positivity of ticks to at least one pathogen was 18%, with variations in the prevalence across the different examined regions, probably due to the optimal temperature and humidity of certain areas [34]. Therefore, while in the north-west of the country the prevalence of positive ticks ranged between 21.4% [35] and 24% [34], in the north-east, an evident lower prevalence (9.4%) was revealed [35]. Nevertheless, the authors concluded that the incidence of human TBDs in Italy is probably underestimated due to the lack of surveillance and low amounts of studies, as well as the need for further studies to better understand the impact of climatic and host factors on the dynamic of ticks and TBDs [34,35]. Considering the above, we have produced a new panel for the search for neglected and not normally researched pathogens of potential zoonotic interest, and also in order to reveal the ecological niches where ticks infected by potentially zoonotic pathogens reside.

Farmed animals that live in close contact with farm workers and, during grazing, with potentially infected ticks of wildlife, may be a useful tool and act as sentinels for the stratification of the risk for public health in order to implement measures for effective management and control in certain habitats. Indeed, the wide percentage of farms testing positive to TBPs herein (32.5%) highlights a not negligible exposure to infections in the livestock industry in Southern Italy. In accordance, the high molecular prevalence for at least one TBP (66.3%), together with the occurrence of co-infections in the animals analyzed, further indicate a broad exposure of the livestock to multiple hard tick species and related pathogens in the study area. However, it is possible to speculate about higher exposure of these animals to the tick-borne pathogens, considering the limitation of molecular techniques to detect pathogenic DNA in the host’s blood, showing very low sensitivity beyond two weeks after infection [36] and mostly in areas that have shown higher risk of being positive. Indeed, the fact that 71% of positive farms are located in particularly humid areas (Figure 1) confirms that humidity represents a crucial environmental parameter for the survival and development of hard ticks [21,37,38]. The higher prevalence of TBPs in farms housing large numbers of animals suggests that a greater abundance of susceptible hosts may also result in a higher frequency of tick infestations and tick bites. The higher positivity observed during May, compared to the other months of sampling (Table 2), is probably due to an overall peak of biological activity and abundance of ticks, especially nymphs and adults, from May to July [21,34,39].

The absence of any clinical signs in the positive animals herein examined suggests an asymptomatic course of the infections, thus making it difficult to suspect the occurrence of these infections in livestock.

From a “*One Health*” perspective, in the present study, the need is underlined for to establishing, possibly in collaboration with livestock operators, a health monitoring plan for the control of TBPs towards domestic animals and humans, in order to prevent the transmission pathway of these infections. Indeed, not surprisingly, the pathogens identified (*A. phagocytophilum*, *Babesia* spp., *C. burnetii* and *B. burgdorferi* s. l complex) have been reported in domestic (hunting dogs—[40]) and wild animals (red foxes, *Vulpes vulpes*—[41], as well as in outdoor workers (farmers, hunters, forestry workers, hikers [20,42]) in the study area. Accordingly, further studies are needed to investigate the tick fauna of livestock in southern Italy, the potential clinical aspects related to infections, and the epidemiology of TBPs where farm, domestic, and wild animals, other than humans, overlap the same environment.

## Figures and Tables

**Figure 1 microorganisms-13-00139-f001:**
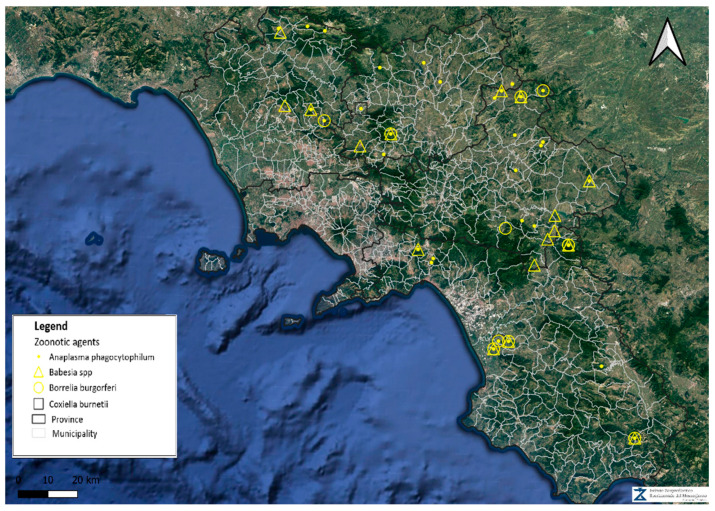
Distribution of the investigated establishments and the tested positive farms for the different pathogens.

**Table 1 microorganisms-13-00139-t001:** Primer sets and probes and thermal profiles for the detection of tick-borne pathogens.

Tick-Borne Pathogen	Primer and Probe Sequence (5′–3′)	Thermal Profile				Ref.
		Step	Temperature (°C)	Time	Cycles (n)	
*A. phagocytophilum*	QAP16sf1-F 5′-TGCCACGGTGAATACGTTCTC-3′	Enzymatic activation	95	2 min		[22]
	QAP16sr1-R 5′-GCGCACCAGCTTCGAGTT-3′	Denaturation	95	5 s	40	
	AP-P- FAM 5′-TACACACTGCCCGTCACGCCATG-3′ BHQ1	Annealing/ Extension	55.5	30 s		
*Babesia* spp.	Bab18S_F 5′-CATGAACGAGGAATGCCTAGTATG-3′	Enzymatic activation	98	3 min		[23]
	Bab18S_R 5′-CCGAATAATTCACCGGATCACTC-3′	Denaturation	95	15 s	40	
	Bab18S_P-FAM 5′-AAGTCATCAGCTTGTGCAGATTACGTCCCT-3′ BHQ1	Annealing/ Extension	60	60 s		
*B. burgdorferi* s. l. complex	Bo_bu_sl_23S_ F 5′-GAGTCTTAAAAGGGCGATTTAGT-3′	Enzymatic activation	98	3 min		[24]
	Bo_bu_sl_23S_ R 5′-CTTCAGCCTGGCCATAAATAG-3′	Denaturation	95	15 s	45	
	Bo_bu_sl_23S_ P-FAM 5′-AGATGTGGTAGACCCGAAGCCGAGT-3′ BHQ1	Annealing/ Extension	60	60 s		
*C. burnetii*	Coxi-F 5′-AAAACGGATAAAAGAGTCTGTGGTT-3′	Enzymatic activation	98	3 min		[25]
	Coxi-R 5′-CCACACAAGCGCGATTCAT-3′	Denaturation	95	15 s	45	
	Coxi P-FAM 5′-AAAGCACTCATTGAGCGCCGCG-3′ BHQ1	Annealing/ Extension	60	60 s		
*Crimean-Congo hemorrhagic fever Virus* (*CCHFV*)	CCHFV-CF-F 5′-CAAGGGGTACCAAGAAAATGAAGAAGGC-3′	Reverse transcription	50°	30 min		[26]
	CCHFV-CR-R 5′-GCCACAGGGATTGTTCCAAAGCAGAC-3′	Enzymatic activation	95	2 min		
	CCHFV-SE01-P-FAM-5′-ATCTACATGCACCCTGCTGCTGTGTTGAACA-3′-TAMRA	Denaturation	95°	15 s	46	
		Annealing/ Extension	59	30 s		
Tick-borne encephalitis virus (TBEV)	TBE-F 5′-TGGAYTTYAGACAGGAAYCAACACA-3′	Reverse transcription	45	15 min		[27]
	TBE-R 5′-TCCAGAGACTYTGRTCDGTGTGGA-3′	Enzymatic activation	95	10 min		
	TBE-P FAM-5′-CCCATCACTCCWGTGTCAC-3′-TAMRA	Denaturation	95	15 s	45	
		Annealing/ Extension	60	60 s		

**Table 2 microorganisms-13-00139-t002:** Prevalence of vector-borne pathogens in cattle/buffalo and sheep/goats in the study area.

Tick-Borne Pathogen	Cattle		Buffalo		Sheep		Goats		Total		95% CI
	(*n* = 310)		(*n* = 35)		(*n* = 220)		(*n* = 56)		(*n* = 621)		
	pos	%	pos	%	pos	%	pos	%	pos	%	
*A. phagocytophilum*	118	38.06	12	34.29	82	37.27	28	50.00	240	38.64	34.9–42.5
*Babesia* spp.	79	25.48	7	20.00	55	25.00	2	3.57	143	23.02	19.9–26.5
*B. burgdorferi* s. l. complex	21	6.77	3	8.57	2	0.91	0	0.00	26	4.18	2.9–6.1
*C. burnetii*	2	0.65	0	0.00	1	0.45	0	0.00	3	0.48	0.2–1.4
TBEV	0	-	0	-	0	-	0	-	0	-	-
CCHFV	0	-	0	-	0	-	0	-	0	-	-
Total	220	70.97	22	62.86	140	63.64	30	53.57	412	66.34	62.5–69.9

**Table 3 microorganisms-13-00139-t003:** Evaluation of different variables in the study area among negative and positive farms to tick-borne pathogens.

Variable	No. Negative (*n* = 88)	No. Positive (*n* = 38)	*p*-Value
**Field context (ground)**			**<0.05**
Dry	41 (46.6%)	7 (18.4%)	
Humid	42 (47.7%)	27 (71.1%)	
**No. of animals housed per farm**			**<0.05**
Mean (SD)	78.4 (160)	140 (149)	
Median [Min, Max]	25.5 [0, 1260]	77.5 [4.0, 496]	
**Sampling month**			**<0.05**
March	48 (54.5%)	9 (23.7%)	
May	6 (6.8%)	18 (47.4%)	
June	12 (13.6%)	6 (15.8%)	
August	22 (25.0%)	5 (13.2%)	

Results were considered as statistically significant with a *p* value < 0.05.

## Data Availability

The original contributions presented in the study are included in the article, further inquiries can be directed to the corresponding author.

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
