# Peer review of "Evaluation of Risk Factors Influencing Tick-Borne Infections in Livestock Through Molecular Analyses"

_microorganisms, 2025, doi:10.3390/microorganisms13010139_

Round 1
Reviewer 1 Report
Comments and Suggestions for Authors
Introduction
-You do not mention examples of tick-borne diseases in livestock
-The aim of this study is not clear. Please, clarify it
Methods
-You have to conduct sequencing of the pathogens detected in this study in some positive samples to show the genetic relatedness with human and animal strains
Discussion
- It should be focus on main findings of the study and correlate them with previous studies
Author Response
Comments to the Author
-You do not mention examples of tick-borne diseases in livestock.
Authors’ response
Examples have been added accordingly.
Comments to the Author
-The aim of this study is not clear. Please, clarify it
Authors’ response
Thank you for the comment. We have modified and better clarified this aspect.
Comments to the Author
-You have to conduct sequencing of the pathogens detected in this study in some positive samples to show the genetic relatedness with human and animal strains
Authors’ response
We agree with this comment. Nevertheless, we preferred to use real-time PCR assays instead of conventional PCR protocols followed by sequencing, in order to guarantee an early-detection of the pathogens. Furthermore, it was frequent the detection of the pathogens by real-time PCR with Ct values over 35, which limited the possibilities to perform any sequencing. However, after the initial screening of the pathogens in this study, we plan to investigate positive samples obtained by the sequencing approach in the future to obtain genetic insights.
Comments to the Author
- It should be focus on main findings of the study and correlate them with previous studies.
Authors’ response
We have integrated in the discussion chapter accordingly with the author request
Reviewer 2 Report
Comments and Suggestions for Authors
Introduction
1. It is essential that the author focuses only on TBD will examine in the manuscript
2. It is important that the author introduces each TBD examined in the manuscript in some detail, rather than providing general information.
Materials and methods
1. In the samples collected by veterinarians, what are the clinical signs of positive animals?
2. It is necessary for the author to separate each animal type with a separate number, not to include cattle and buffalo together.
3. There is a need for the author to distinguish between sheep and goats
4. The location of the samples in figure one is unclear.
5. Why is the author focusing only on this TBD out of all TBDs
Table 1
2. The author needs to add the abbreviation under the table for each abbreviation used in the table
3. What are the conditions for PCR reaction for each primer, all the conditions need to be found in separate tables.
Result
Table 2
1. Data and numbers for cattle and boffola should be separated
2. A separation between sheep and goat is needed by the author
Discussion
It is important to focus on what is novel in the manuscript and why the author should limit themselves to this type of TBD disease rather than other types of TBD diseases
It is important for the author to discuss each result separately and reflect on its significance to public health.
Comments on the Quality of English Language
Quality of English could be improved to more clearly express the research
Author Response
Comments to the Author
- It is essential that the author focuses only on TBD will examine in the manuscript
Authors’ response
The introduction section has been amended accordingly.
Comments to the Author
- It is important that the author introduces each TBD examined in the manuscript in some detail, rather than providing general information.
Authors’ response
Details on each TBD examined in the manuscript have been added.
Comments to the Author
- In the samples collected by veterinarians, what are the clinical signs of positive animals?
Authors’ response
All the animals in this study displayed an apparent good health status, as the local health authorities did not report any evident clinical signs, despite the positivity to several tick-borne pathogens.
Comments to the Author
- It is necessary for the author to separate each animal type with a separate number, not to include cattle and buffalo together.
Authors’ response
We agree with this comment, therefore, we have separated the animals as requested.
Comments to the Author
- There is a need for the author to distinguish between sheep and goats
Authors’ response
Corrected, accordingly as the previous comment.
Comments to the Author
- The location of the samples in figure one is unclear.
Authors’ response
We have changed the map and better represent the study area
Comments to the Author
- Why is the author focusing only on this TBD out of all TBDs
Authors’ response
As reported by the Italian National Health Institute (ISS), the main TBDs that show higher epidemiological significance for humans in Italy are tick-borne encephalitis, Lyme disease, rickettsiosis, tick-borne relapsing fever, tularemia, tick-borne meningoencephalitis. Furthermore, we decided to include TBDs that create concern for the introduction among neighbour countries. Nevertheless, despite the health implications, many tick-borne zoonoses are not monitored and often poorly investigated, We preferred to focus on TBDs common in the study areas or those considered emerging.
Comments to the Author
- The author needs to add the abbreviation under the table for each abbreviation used in the table
Authors’ response
Done.
Comments to the Author
- What are the conditions for PCR reaction for each primer, all the conditions need to be found in separate tables.
Authors’ response
Done.
Comments to the Author
- Data and numbers for cattle and boffola should be separated
Authors’ response
Done.
Comments to the Author
- A separation between sheep and goat is needed by the author
Authors’ response
Done.
Comments to the Author
It is important to focus on what is novel in the manuscript and why the author should limit themselves to this type of TBD disease rather than other types of TBD diseases
Authors’ response
Thank you for the comment. Neglected TBD’s of farming interest were focused as the constructed panel was intended to reveal the ecological niches where ticks infected by potentially zoonotic pathogens reside. We have integrated the discussion with the relative comment.
Comments to the Author
It is important for the author to discuss each result separately and reflect on its significance to public health.
Authors’ response
We have underlined the significance in public health issues into the discussion chapter.
Reviewer 3 Report
Comments and Suggestions for Authors
The manuscript explores the prevalence of tick-borne pathogens (TBPs) in livestock species (sheep, goats, cattle, and buffaloes) in southern Italy, focusing on potential environmental risk factors associated with their transmission.
While the introduction provides a solid context, the primary objective of the study is not clearly defined. Clarifying whether the main aim is to identify risk factors, measure exposure, or provide guidance for surveillance would improve the study's coherence and enhance reader understanding. Additionally, although the manuscript provides a general overview of tick-borne disease distribution in Italy, it would be helpful to contextualize the study by including a comparison with previous studies in similar regions. This would highlight gaps in existing research and justify the necessity of the present study.
The study focuses on tick-borne pathogens but does not present data on tick populations or infestation density on animals. Although pathogens were tested in animal blood samples, it is unclear whether there was a direct correlation between tick density and pathogen prevalence. Including data on tick populations would facilitate a direct correlation between tick density, tick species diversity, and pathogen prevalence.
The manuscript mentions that no clinical signs were observed in the animals but does not provide further details on the clinical examination process. Since some tick-borne infections can be asymptomatic or cause subclinical infections, the absence of clinical signs should be more thoroughly explained. Moreover, the study could benefit from an investigation into potential indicators of infection, such as changes in productivity or milk yield among the animals.
Information on the handling of confounding factors, such as differences in animal management practices, tick control measures, or seasonal variations in vector activity, would be valuable. Although some environmental factors are examined, there is a lack of detailed information on data collection and analysis for these variables. How were these factors measured? Were specific tools used to monitor temperature and humidity, or were they derived from secondary sources (e.g., weather stations)? The methodology for collecting and managing environmental data should be explained in greater detail.
While the sample of 621 animals from 126 farms appears ample, it is not discussed whether the farms were selected randomly or if there was potential selection bias.
High prevalences of co-infection are reported; however, the manuscript does not discuss the clinical and epidemiological implications of these co-infections in detail. Additionally, no specific information is provided on the geographical distribution of pathogens. Given that the study was conducted on farms distributed across various provinces in Campania, it would be interesting to explore potential differences between more humid or mountainous areas and drier regions. Geographic data, such as mapping of infected areas, could add significant value to the analysis of environmental factors.
Author Response
Comments to the Author
The manuscript explores the prevalence of tick-borne pathogens (TBPs) in livestock species (sheep, goats, cattle, and buffaloes) in southern Italy, focusing on potential environmental risk factors associated with their transmission.
While the introduction provides a solid context, the primary objective of the study is not clearly defined. Clarifying whether the main aim is to identify risk factors, measure exposure, or provide guidance for surveillance would improve the study's coherence and enhance reader understanding.
Authors’ response
Thank you for the comment. The main objective of the study was to assess the prevalence of selected TBPs in farm animals and potential risks of pathogen transmission. We hope to have better address the manuscript through discussion chapter.
Comments to the Author
Additionally, although the manuscript provides a general overview of tick-borne disease distribution in Italy, it would be helpful to contextualize the study by including a comparison with previous studies in similar regions. This would highlight gaps in existing research and justify the necessity of the present study.
Authors’ response
We have consider this aspect and we have researched for similar data in order to compare the study with other regions. Unfortunately we have found only old papers described with different methods. Anyway we have integrated the discussion with further data.
Comments to the Author
The study focuses on tick-borne pathogens but does not present data on tick populations or infestation density on animals. Although pathogens were tested in animal blood samples, it is unclear whether there was a direct correlation between tick density and pathogen prevalence. Including data on tick populations would facilitate a direct correlation between tick density, tick species diversity, and pathogen prevalence.
Authors’ response
Thank you for the relevant comment. Actually, We are conducting further study in order to establish the ticks abundance and related pathogens prevalence in the same sites in which we have revealed the presence in farmed animals. In this study, the presence and abundance of ticks were not yet investigated as we are building further aimed related studies. Furthermore no official data are available.
Comments to the Author
The manuscript mentions that no clinical signs were observed in the animals but does not provide further details on the clinical examination process. Since some tick-borne infections can be asymptomatic or cause subclinical infections, the absence of clinical signs should be more thoroughly explained. Moreover, the study could benefit from an investigation into potential indicators of infection, such as changes in productivity or milk yield among the animals.
Authors’ response
All the animals in this study displayed an apparent good health status, despite the positivity to several tick-borne pathogens. This study is an initial screening for assessing the circulation of TBPs in farm animals, but our future surveys will explore potential implications of these infections, such as clinical and productive aspects.
Comments to the Author
Information on the handling of confounding factors, such as differences in animal management practices, tick control measures, or seasonal variations in vector activity, would be valuable. Although some environmental factors are examined, there is a lack of detailed information on data collection and analysis for these variables. How were these factors measured? Were specific tools used to monitor temperature and humidity, or were they derived from secondary sources (e.g., weather stations)? The methodology for collecting and managing environmental data should be explained in greater detail.
Authors’ response
Thank you. Details added in M&M chapter
Comments to the Author
While the sample of 621 animals from 126 farms appears ample, it is not discussed whether the farms were selected randomly or if there was potential selection bias.
Authors’ response
Details added in M&M chapter as the previous requested comment.
Comments to the Author
High prevalences of co-infection are reported; however, the manuscript does not discuss the clinical and epidemiological implications of these co-infections in detail. Additionally, no specific information is provided on the geographical distribution of pathogens. Given that the study was conducted on farms distributed across various provinces in Campania, it would be interesting to explore potential differences between more humid or mountainous areas and drier regions. Geographic data, such as mapping of infected areas, could add significant value to the analysis of environmental factors.
Authors’ response
Being a preliminary survey, we did not collect detailed climatic information, but future surveys will investigate potential environmental aspects of the different provinces of the study area related to the pathogen occurrence.
Reviewer 4 Report
Comments and Suggestions for Authors
Abstract:
o The abstract effectively summarizes the study’s findings but could benefit from clarifying the scope (i.e., the livestock types or geographic focus) to provide readers with immediate context.
Introduction:
o The introduction provides a solid background on vector-borne diseases (VBDs) and the relevance of ticks in their transmission.
o While it’s informative, consider briefly stating the study objectives or the specific hypotheses being tested towards the end to strengthen its purpose and flow.
Materials and Methods
o You mentioned that you used a positive control, but please specify whether you used a synthetic or previously generated sample.
o If you describe each PCR condition, it would be helpful to other researchers.
Discussion:
o The discussion is thorough, contextualizing the results with existing literature and emphasizing the impact of environmental conditions on tick-borne pathogen prevalence.
o It would benefit from further discussion on limitations, such as any sample biases or PCR sensitivity constraints. This transparency can add credibility to the findings.
o Mentioning future research directions, especially on possible interventions for reducing tick-borne pathogen transmission, could provide a practical perspective on the study’s implications.
Conclusion:
o The conclusion is concise, though it may be strengthened by reiterating the study's broader relevance to public health or livestock management practices in tick-prone regions.
Author Response
Comments to the Author
The abstract effectively summarizes the study’s findings but could benefit from clarifying the scope (i.e., the livestock types or geographic focus) to provide readers with immediate context.
Authors’ response
The main objective of the study was to provide data on the circulation of TBPs in livestock, which are scantly available in the literature in the study area. For the sake of clarity, the abstract was modified accordingly.
Comments to the Author
The introduction provides a solid background on vector-borne diseases (VBDs) and the relevance of ticks in their transmission.
While it’s informative, consider briefly stating the study objectives or the specific hypotheses being tested towards the end to strengthen its purpose and flow.
Authors’ response
We rephrased the introduction section accordingly.
Comments to the Author
You mentioned that you used a positive control, but please specify whether you used a synthetic or previously generated sample.
Authors’ response
All the positive controls used herein were field samples obtained previously generated from other studies.
Comments to the Author
If you describe each PCR condition, it would be helpful to other researchers.
Authors’ response
Done.
Comments to the Author
The discussion is thorough, contextualizing the results with existing literature and emphasizing the impact of environmental conditions on tick-borne pathogen prevalence.
Authors’ response
Done
Comments to the Author
It would benefit from further discussion on limitations, such as any sample biases or PCR sensitivity constraints. This transparency can add credibility to the findings.
Authors’ response
Thank you for the suggestion. We added this consideration in results chapter.
Comments to the Author
Mentioning future research directions, especially on possible interventions for reducing tick-borne pathogen transmission, could provide a practical perspective on the study’s implications.
Authors’ response
Done.
Comments to the Author
The conclusion is concise, though it may be strengthened by reiterating the study's broader relevance to public health or livestock management practices in tick-prone regions.
Authors’ response
Done.
Round 2
Reviewer 1 Report
Comments and Suggestions for Authors
The number of evaluated risk factors is low in this study as well as you conducted only real-time PCR.
You provide simple results that lack innovation in this study
Reviewer 2 Report
Comments and Suggestions for Authors
The author did most of the coments
Comments on the Quality of English LanguageEnglish quality is good
Reviewer 3 Report
Comments and Suggestions for Authors
The authors have made revisions to the manuscript, and for many observations, they have referred to future studies. However, satisfactory improvements have been made.
Reviewer 4 Report
Comments and Suggestions for Authors
The current revised version has been well revised to reflect the points pointed out. Thank you for your hard work.